# Antimicrobial Peptides in Early-Life Host Defense, Perinatal Infections, and Necrotizing Enterocolitis—An Update

**DOI:** 10.3390/jcm11175074

**Published:** 2022-08-29

**Authors:** Eleni Agakidou, Charalampos Agakidis, Angeliki Kontou, William Chotas, Kosmas Sarafidis

**Affiliations:** 11st Department of Neonatology and Neonatal Intensive Care, School of Medicine, Faculty of Health Sciences, Aristotle University of Thessaloniki, Ippokration General Hospital, 49 Konstantinoupoleos Str, 54642 Thessaloniki, Greece; 21st Department of Pediatrics, School of Medicine, Faculty of Health Sciences, Aristotle University of Thessaloniki, Ippokration General Hospital, 49 Konstantinoupoleos Str, 54642 Thessaloniki, Greece; 3Department of Neonatology, University of Vermont, Burlington, VT 05405, USA

**Keywords:** antiproteases, defensins, chorioamnionitis, fetus, hepcidin, host defense peptides, lactoferrin, LL-37, neonate, sepsis

## Abstract

Host defense against early-life infections such as chorioamnionitis, neonatal sepsis, or necrotizing enterocolitis (NEC) relies primarily on innate immunity, in which antimicrobial peptides (AMPs) play a major role. AMPs that are important for the fetus and neonate include α and β defensins, cathelicidin LL-37, antiproteases (elafin, SLPI), and hepcidin. They can be produced by the fetus or neonate, the placenta, chorioamniotic membranes, recruited neutrophils, and milk-protein ingestion or proteolysis. They possess antimicrobial, immunomodulating, inflammation-regulating, and tissue-repairing properties. AMPs are expressed as early as the 13th week and increase progressively through gestation. Limited studies are available on AMP expression and levels in the fetus and neonate. Nevertheless, existing evidence supports the role of AMPs in pathogenesis of chorioamnionitis, neonatal sepsis, and NEC, and their association with disease severity. This suggests a potential role of AMPs in diagnosis, prevention, prognosis, and treatment of sepsis and NEC. Herein, we present an overview of the antimicrobial and immunomodulating properties of human AMPs, their sources in the intrauterine environment, fetus, and neonate, and their changes during pre- and post-natal infections and NEC. We also discuss emerging data regarding the potential utility of AMPs in early-life infections, as diagnostic or predictive biomarkers and as therapeutic alternatives or adjuncts to antibiotic therapy considering the increase of antibiotic resistance in neonatal intensive care units.

## 1. Introduction

Perinatal infections and necrotizing enterocolitis (NEC) have been associated with significant morbidity, mortality, and adverse long-term consequences in neonates. The prevalence of complications in survivors is inversely correlated with gestational age, affecting around 20% of infants weighing less than 1500 g at birth and up to 60% of those with birth weights less than 1000 g [1,2]. Immaturity of the immune system, especially in preterm neonates, is an important factor contributing to increased vulnerability to infections during the perinatal and neonatal period. In early life, host defense relies primarily on the innate immune system, as full function of the acquired immunity has not yet been established due to limited intrauterine exposure to antigens. Innate immunity, which is the first line of defense against invading microorganisms, includes epithelial barriers (skin, mucosal surfaces), immune cells (neutrophils, monocytes, dendritic cells, etc.), inflammatory mediators, antimicrobial proteins (such as lysozyme, azurocidin, cathepsin G, phospholipase A2, and lactoferrin), and antimicrobial peptides (AMPs) [3].

AMPs are small peptides, mostly cationic, containing 7–100 amino acids. However, AMPs with a higher number of amino acids have also been reported [4,5,6]. AMPs are either naturally produced by living organisms, or synthetically produced by modification of natural AMPs. Natural AMPs are found abundantly in nature, where they are produced by plants, microorganisms, fish, amphibians, insects, and mammals, either constitutively or in response to infectious and inflammatory stimuli [7], and exert potent antimicrobial activity against bacteria, fungi, viruses, and protozoa [8]. Compared to other immune components such as immunoglobulins, one advantage of AMPs is their ability to be produced rapidly at the expense of relatively low energy. Furthermore, their short amino acid chain facilitates a rapid approach to the target and effective diffusion through the microbial cell membrane [9,10].

The main mechanism of AMP antimicrobial action is the disruption of the cell membrane [10]. This action is mediated by hydrophobic amino acids and cationic charge, which facilitate the binding of AMPs to the cell membrane of microorganisms, (composed of anionic lipids—phosphatidylglycerol and cardiolipin) [11]. Following their attachment to the cell membrane, the AMPs form pores allowing the outflow of anions through the membrane, leading to membrane depolarizing, thus killing the bacteria [12]. In contrast, AMPs have low affinity to the mammalian cell membrane, which is rich in electrically neutral phospholipids (phosphatidylcholine and sphingomyelin) [13]. For certain AMPs, non-membrane mechanisms of antibacterial action have also been described, including entering the bacterial cell after binding to the cell surface receptors [6,8]. Within the cell, AMPs inhibit critical cell functions such as DNA replication, protein synthesis and folding, protease activity, cell division, and cell metabolism [6,8] (Figure 1).

AMPs are also known as host defense peptides, as they mediate many immune and inflammatory responses. In addition, they are involved in certain biochemical and metabolic processes, including promotion of neovascularization, wound healing, and neutralization of bacterial toxins [10,12,14,15]. In this review, we present an overview of the antibacterial and immunomodulating properties of the human AMP families, their sources during intrauterine and early postnatal life, and current data regarding their changes during fetal or neonatal infections and NEC. In addition, we present emerging data on the potential utility of AMPs as diagnostic and predictive biomarkers of perinatal infections and NEC. Finally, given the increase in antimicrobial resistance in neonatal intensive care units, we briefly discuss the challenge of identifying and developing new synthetic analogs of AMPs as alternatives or adjuncts to antibiotic therapy.

## 2. Families and Functions of Human AMPs

Of the 2619 AMPs registered in the Antimicrobial Peptide Database 3 (APD3), 112 are found in *Homo sapiens* [16]. Human AMPs that are clinically important in early life belong mainly to the defensin and cathelicidin families. Additional AMPs outside these families include the antileukoproteases elafin and secretory leukocyte peptidase inhibitor (SLPI), hepcidin, and human milk protein-derived AMPs [4,17,18,19].

### 2.1. Defensins and Cathelicidin LL-37

#### 2.1.1. Structure, Regulation, and Sources of Human Defensins

Defensins are cationic AMPs containing 18–45 amino acid residues. According to their size and the arrangement of the disulfide bonds, they are divided into three subfamilies; alpha (α)-defensins, beta (β)-defensins, and theta (θ)-defensins [4,10,20]. In humans, only α- and β-defensins have been identified [20,21,22].

The α-defensins are AMPs comprising 29–35 amino acid residues, which are activated after proteolytic cleavage of the C-terminal region from a pro-peptide following microbial stimulation [23]. In humans, six α-defensins have been described; the human neutrophil peptides (HNPs) -1,-2,-3, and -4, and human defensin (HD)-5 and HD-6 [23]. Of the α-defensins, HNP-1, -2, -3, and -4 are produced by neutrophils as processed peptides and are abundant in the neutrophil azurophilic granules and phagocytic vacuoles, constituting 99% of the total neutrophil defensins. HD-5 and HD-6 are produced and stored as pro-peptides in the secretory vesicles of the Paneth cells, while HD-5 is also expressed by multiple epithelia of the female reproductive tract, placenta, and fetal membranes (Table 1) [20,24,25,26,27,28].

The β-defensins are synthesized as 64–68 amino acid pre-propeptides, which are eventually activated to the mature β-defensins containing about 35 amino acid residues. Four β-defensins have been isolated from humans; the human β-defensins (HBD)-1 to -4 [4,29], expressed in neutrophils and other immune cells. These are the predominant defensins isolated from keratinocytes and epithelial cells of the respiratory, gastrointestinal (GI), and genitourinary tracts, representing the first-line barrier against environmental threats [29,30]. Moreover, β-defensins can be found in blood, urine, heart, skeletal muscles, and testes [8,27,31]. HBD-1 and -2 are constitutively expressed, while the production of HBD-2 is also induced by pro-inflammatory cytokines, microbial lipopolysaccharide (LPS), bacteria, and fungi (Table 1) [29,32,33,34].

#### 2.1.2. Structure, Regulation, and Sources of Human Cathelicidin LL-37

Cathelicidins are amphipathic, positively charged peptides encoded in humans by the cathelicidin antimicrobial peptide *(CAMP)* gene. *CAMP* encodes the precursor cationic antimicrobial protein (CAP)-18 (18 kDa), which undergoes proteolysis by proteinase-3 to release two peptides: the cathelin and the LL-37 [35]. Both of these cathelicidin-derived peptides exert antimicrobial properties, albeit only the latter is found in humans. In humans, LL-37 is produced by a wide range of tissues, including skin, lung, and GI epithelial cells, cervix, vagina, and immune cells (neutrophils, monocytes, macrophages, dendritic cells, mast cells, and platelets), and is released in plasma, bronchoalveolar and gastric fluids, breast milk, saliva, and other biofluids (Table 1) [23,36]. LL-37 is constitutionally expressed in certain cells (e.g., sweat glands), and its synthesis in other cells (e.g., keratinocytes) is induced following activation by infectious agents and vitamin D [37].

#### 2.1.3. Antimicrobial Actions of Defensins and LL-37

Human defensins and LL-37 are effective against a broad spectrum of microorganisms including gram-positive and gram-negative bacteria, viruses, and fungi, as indicated in Table 1 [20,30,34,38,39,40,41,42,43].

Several studies have shown that LL-37 acts synergistically with HNP-1, HBD-2, and HBD-3 against pathogens that cause neonatal infections [42,44,45]. Promotion of gut colonization by beneficial bacteria to prevent antibiotic-induced dysbiosis is another important function of α-defensins [46]. In addition, LL-37 increases the effectiveness of antibiotics by inhibiting biofilm formation [47,48].

#### 2.1.4. Immunomodulating Properties of Defensins and LL-37

In addition to their bactericidal activities, defensins and LL-37 have important immunomodulating and inflammation-regulating properties, as summarized in Table 1 and Figure 2 [13,31,45,46,49,50,51,52,53,54,55,56]. They exert a strong chemotactic action on immune cells, and regulate cytokine, chemokine, and adhesion molecule production [34,38,40,51,57,58,59,60]. In vitro and experimental studies have shown that LL-37 suppresses the pyroptosis and activation of macrophages, as well as the development of neutrophil apoptosis, while stimulating bactericidal capacity [13,50,53,54,61]. In addition, HBD-1 promotes the differentiation of cord-blood monocytes to immature dendritic cells, and inhibits their apoptosis. HBD-3 exerts mainly anti-inflammatory actions [50].

The regulating effects of defensins and LL-37 on inflammation and immune response indicate a bi-directional action of these AMPs at the site of infection, protecting tissues from potentially excessive inflammatory response [49,50,62]. Moreover, human defensins have been reported to induce the release of histamine and prostaglandin D2 (PGD2) through activation of mast cells [63], and to inhibit the activation of the classical complement pathway [64]. Further functions of β-defensins and LL-37 are important for the preservation of epithelial barrier integrity, as well as the amelioration and repair of inflammation-induced tissue injury. In this context, they not only enhance angiogenesis, cell migration, and cell proliferation, but also exert antioxidant action, promote re-epithelization, and strengthen the tight junctions in mucosal and skin epithelia [50,65,66].

Importantly, defensins also act as inducers of the acquired immune system by activating and recruiting immune cells, thereby linking innate with acquired immunity [58,67].

In conclusion, substantial evidence accumulated in recent years indicates that human defensins and LL-37 are multifunctional, and are involved in both innate and adaptive immunity by interacting with host-cell receptors. A better understanding of the role of defensins and LL-37 in immune response may have implications for potential clinical therapeutic use against infections.

### 2.2. Antileukoproteases Elafin and Secretory Leukocyte Protease Inhibitor

Antileukoproteases are a family of AMPs including two members, elafin (skin-derived antiprotease) and SLPI [68]. Elafin is a peptide of 95 amino acids, and SLPI has 107 amino acids. Both antileukoproteases are constitutively produced by epithelia in the skin [69], respiratory tract [70,71], and intestine [72], as well as in the amniotic membrane, endometrium [73], neutrophils, and macrophages (Table 1) [70,73,74,75,76]. The synthesis and secretion of SLPI and elafin increase at the site of inflammation in response to IL-1, TNF-α, LPS, α-defensins, and hormones such as progesterone [70,75].

#### Functions of Antileukoproteases

The main function of antiproteases is to neutralize proteases derived from inflammatory cells, thus preventing tissue damage caused by excessive protease activity [77]. In addition to their role as antiproteases, SLPI and elafin possess antibacterial and anti-inflammatory properties and regulate innate and acquired immune responses, favoring a shift towards a Th1 immune response [69,78]. The exact antimicrobial mechanisms related to elafin and SLPI have not yet been fully elucidated. However, the cationic nature of these two peptides suggests a membrane-associated antimicrobial mechanism similar to that of other AMPs (Figure 1). Elafin has been found to be effective against certain gram-positive and -negative bacteria, as well as fungi (Table 1) [77,79,80].

### 2.3. Hepcidin

Human hepcidin, also known as HAMP (human anti-microbial peptide), is an amphipathic peptide involved in iron homeostasis by controlling intestinal iron absorption and storage in macrophages. In addition, hepcidin plays an important role in innate immunity due to its antimicrobial properties. The bioactive form of mature hepcidin is a 60-amino-acid peptide produced in the liver after proteolytic cleavage of pro-hepcidin [81]. Due to the positively charged amphipathic nature of hepcidin, its antibacterial effect is cell-membrane-dependent, similar to that of defensins and LL-37. It is produced by hepatocytes, neutrophils, and macrophages. In vitro and in vivo experimental studies have shown that hepcidin mRNA expression was upregulated in response to LPS and inflammatory cytokines (IL-6, IL-1α and IL-1β), and inhibited by anti-IL-6 antibodies [18,82,83,84]. Besides its direct bactericidal effect, hepcidin indirectly inhibits bacterial growth through its regulating effect on iron metabolism, as the hepcidin-induced hypoferremia during inflammation deprives microbes of the iron needed for their growth. Furthermore, hepcidin acts as an acute-phase protein in response to inflammation and hypoxia, and protects against LPS-induced endotoxemia [85].

### 2.4. Human Milk-Derived AMPs

Human milk is an important source of AMPs that play a central role in hosts’ innate intestinal defense and in the development of NEC, especially form preterm neonates who are at high risk for both sepsis and NEC [86,87]. Several AMPs are produced in the gut following proteolysis and the fermentation of milk proteins (e.g., casein, a-lactalbumin, lactoferrin, and lysozyme; see Table 2) [17,86,87,88,89,90,91]. Additionally, HNP-1 and -2, HD-5 and HD-6, HBD-1 to -4, and LL-37 have been identified in human milk and colostrum [36,86,87,92].

## 3. AMP Sources in the Intrauterine Environment, the Fetus, and the Neonate

### 3.1. AMPs Originating from the Maternal Reproductive Tract during Pregnancy

Although the intrauterine environment was for decades thought to be sterile, numerous studies have since documented the presence of microorganisms in the amniotic fluid (AF) and meconium, even in the absence of fetal membrane rupture [93]. Actually, studies using omics and novel molecular techniques have revealed that normal microflora and pathogens present in placentas, chorioamniotic membranes, and AF [94,95]. However, currently it is not clear as to whether the microbes detected in the amniotic cavity represent microbial colonization or contamination [96,97]. Nevertheless, the risk of fetal and neonatal short- and long-term morbidities associated with prenatal exposure to microbes and/or inflammation is a source of major concern for obstetricians and neonatologists [96,98]. The fetomaternal unit possesses several anti-infective mechanisms, overall constituting the intrauterine innate host defense system. These mechanisms contribute significantly to the protection of the fetus by creating an anatomic barrier against microbe invasion and by producing antimicrobial and immunomodulating factors, including AMPs. Sources of AMPs in the maternal reproductive system during pregnancy are shown in Figure 3. In addition, AMPs have been detected in AF, cervical mucus plugs, vernix caseosa, and cord blood. These components exhibit antibacterial actions that are partly attributed to the presence of α- and β-defensins, LL-37, SLPI, and elafin [28,99,100,101,102,103,104,105,106,107,108,109,110,111,112,113,114,115,116,117].

Existing data suggests that HNPs 1–3 and HD-5 are produced by the female reproductive system during pregnancy (Figure 3) and they have been detected in the AF of uncomplicated and complicated pregnancies [26,28,101,102,103,104,105,106,115]. HBDs are among the most important AMPs during fetal and early postnatal life. They are constitutively expressed in fetal membranes and placenta tissue, and are upregulated by inflammatory cytokines [106,118,119,120,121,122]. HBD expression level is positively correlated with antimicrobial activity of the membranes. The highest expression has been observed for HBD-3, which is a dominant defensin in the amniotic epithelium. In vitro studies on cultured fetal membranes showed increased production of HNP-1 to -3 and HBD-1 and -2 in the presence of *E. coli* infection [123].

Cathelicidin LL-37 levels remain relatively unchanged throughout uncomplicated pregnancies (Figure 3) [124,125]. Some researchers failed to demonstrate the presence of either HBD 2 or LL-37 in the AF of uncomplicated pregnancies, probably because the expression of these AMPs is inducible [106,123].

Elafin and SLPI are produced by the epithelia of the maternal reproductive system and by inflammatory cells (Table 2, Figure 3) [69,74,118,120,126,127]. SLPI serum levels decrease in early pregnancy and return to non-pregnant values intrapartum. Levels in cervical mucus increase significantly during pregnancy and remain elevated after delivery [128]. Production of elafin and/or SLPI increases in response to microbial products (LPS), cytokines (IL-1, TNF-α), and α-defensins [69,76,116,123,129].

Overall, the presence of numerous AMPs in normal pregnancy and their induction during intra-amniotic infections highlight their important role in the prevention and resolution of intrauterine infection and inflammation.

### 3.2. AMP Production by the Fetus and Neonate

Several studies have shown that the fetus and neonate are capable of synthesizing α- and β-defensins, LL-37, elafin, and SLPI (Figure 3) [107,130]. Skin, respiratory, and GI epithelium as well as blood cells are the main sources of fetal and neonatal AMPs. Clinical studies regarding AMP expression and levels in the fetus and neonate and changes during neonatal infections and NEC are summarized in Table 3.

#### 3.2.1. The Fetal and Neonatal Skin as a Source of AMPs

The skin serves as a mechanical and immunological barrier to environmental intruders [157]. However, fetal and neonatal skin is very thin and is vulnerable to injuries, which along with high postnatal exposure to pathogens, increases the risk of infections, especially in preterm neonates. To mitigate these risks, the neonatal skin contains inflammatory cells and AMPs produced by keratinocytes, normal squamous epithelia, sebocytes, mast cells, and locally recruited immune cells. LL-37 and β-defensins are the best defined AMPs in neonatal skin, although SLPI and elafin have been also detected (Figure 3) [129].

The expression of certain AMPs, such as β-defensins, LL-37, and elafin, is low or absent in the normal skin of adults, but can be induced by inflammation [158]. In contrast, several studies in fetal and neonatal keratinocytes, skin, and vernix caseosa have documented the ability of fetal skin to constitutively synthesize AMPs [129,141,150]. More specifically, as shown by early studies, the skins of mice embryos and newborns and also the foreskins of human neonates were found to express cathelicidins and β-defensins at significantly higher levels than adult skin [150]. These findings were confirmed by in vitro studies on human keratinocytes. It was found that keratinocytes from fetuses with 22–23 weeks of gestation expressed HBD-2, HBD-3, and LL-37 at levels significantly higher than keratinocytes from neonatal foreskin or adult skin. Furthermore, the production of HBD-2 and -3, but not HBD-1 and LL-37, was induced with the stimulation of pre- and post-natal keratinocytes by toll-like receptor (TLR)-signaling and cytokines [151]. Likewise, a dense expression of LL-37 was found in skin lesions of neonates with erythema toxicum. Interestingly, both LL-37 and HBD-1 were constitutionally expressed throughout the entire epidermal layer even in infants without rash [148]. Other studies reported the detection of elafin expression in the developing epidermis of the fetus from the 28th week of gestation up to the first month postpartum, and in the keratinocytes of tongue epithelium from as early as the 17th week of gestation until adult life [129]. In summary, several studies have shown that the fetal and neonatal skin is able to constitutively produce β-defensins, LL-37, HNP-1 to -3, elafin, and SLPI. Moreover, AMP expression, mainly of HBD-2, is upregulated in response to inflammatory and infectious stimuli.

In addition, the fetus and neonate are protected by the vernix caseosa, a lipid-rich material covering their skin from the third trimester onward up to the first week of life, if it is not removed after birth [30,159]. The vernix caseosa provides protection from infections through its physical and anti-infective properties. It contains several antimicrobial peptides and proteins including LL-37, HNP-1 to -3, SLPI, lactoferrin, lysozyme, and calprotectin, and was found to inhibit the proliferation of group B *Streptococcus*, and of *K. pneumoniae* and *Listeria monocytogenes* [105,106,141,148,159]. Vernix production by the fetal sebaceous glands begins at 20 weeks of gestation, and the low amount of vernix in preterm infants born prior to the 28th week of gestation contributes to their increased risk of infections [30].

#### 3.2.2. The Developing Lung as a Source of AMPs

The extensive surface of respiratory epithelium is at risk bacterial contamination and invasion. Therefore, the airway epithelium produces multiple substances with antibacterial and immunomodulating functions, including AMPs [160]. In vitro studies as well as studies on fetal lungs and the bronchoalveolar fluid (BAF) of ventilated neonates have documented the production of AMPs in the developing human lungs by epithelial cells, alveolar macrophages, and recruited immune cells. The AMPs produced in the lung include α- and β-defensins, LL-37, elafin, and SLPI [70,75,109,161]. The HBD-2 is the predominant AMP in the lung [109], and is expressed in the pulmonary epithelium in higher concentrations than in other tissues [32]. The developmental changes of the HBD and LL-37 expression after birth were assessed in cultures of epithelial cells from tracheal aspirates of adults and intubated neonates, as well as in fetal lung explant cultures. HBD-2 mRNA was found to be expressed in lung tissues at term and in two postnatal samples (seven months and 13 years of age), but not in the prenatal lung tissues (18 and 22 gestational weeks). HBD-1 was also expressed, albeit at a lower degree, while HBD-3 was not expressed. Of note, LL-37 mRNA was expressed in all tissue samples with no apparent developmental changes [109,149]. HNPs-1 to -4 have also been detected in BAF and were positively correlated to the numbers of neutrophils [111]. HBD-2 mRNA expression was induced by IL-1β and downregulated by dexamethasone [109]. The antiproteases elafin and SLPI are constitutively produced by the lung epithelial cells, macrophages, and neutrophils, and have been detected in human lung secretions [162]. Beyond their effect on preventing lung infections, antiproteases may play a role in preventing ventilator- and oxygen-induced lung injury, which potentially leads to bronchopulmonary dysplasia [163,164]. This fact is supported by early studies in mechanically ventilated preterm neonates, which reported low levels of SLPI in BAF of neonates with respiratory distress syndrome [152]. Moreover, an imbalance between elastase and SLPI activity (with the presence of more elastase) has been associated with the development of ventilator- or hyperoxia-induced lung injury [153].

In summary, available data show that human α- and β-defensins, LL-37, and antiproteases are expressed in developing lung epithelial cells, macrophages, and recruited neutrophils. They have been detected in BAF, with HDB-2 being the predominant AMP.

#### 3.2.3. The Fetal and Neonatal Gastrointestinal System as a Source of AMPs

The GI system has the largest surface area in the human body and is the entrance point of many microorganisms. As the first line of defense against pathogens, the GI tract needs a fully functioning innate immunity. Data on AMP expression in the GI systems of healthy human neonates are almost entirely lacking. Available data only refer to AMP expression in animals, feces, GI tissues from fetuses, and from neonates with NEC. The AMPs produced in the gut include α-defensins HD-5 and HD-6, HNP-1 to -4, β-defensins HBD-1 to -3, cathelicidin LL-37, and antiproteases. Moreover, the neonatal intestine is protected by AMPs derived from human milk, which are especially important for preterm neonates, who are at high risk of NEC development.

The Paneth cells located in the crypts of Lieberkühn are critical in maintaining the integrity of the intestinal barrier and homeostasis of the small intestine. Paneth cells first appear in the colon and small intestine at 13.5 weeks of gestation, and are present in the small intestine only after the 17th week of gestation [132,165,166,167]. Their density and functionality progressively increase throughout gestation (primarily during the third trimester), but do not reach adult levels before term [135,168]. The Paneth cells produce α-defensins HD-5 and HD-6, constitutionally and in response to infectious stimuli [165]. While α-defensins can first be detected in the Paneth cells at 17 weeks of gestation, α-defensin mRNA can only be measured after the 19th–24th week, at levels approximately 40- to 250-fold lower than those found in adults [127,132,133]. Overall, the Paneth-cell-derived α-defensins in the gut are the predominant antimicrobial factors against enteric pathogens. [133,169].

Human β-defensins are constitutionally expressed in epithelial cells throughout the GI tract, with HBD-1 and HBD-2 being the most extensively studied GI β-defensins [29]. Longitudinal assessment of HBD-1 and HBD-2 levels in the econium and feces of healthy extremely-low-birth-weight infants demonstrated a significant decrease in both β-defensins during the first postnatal weeks, followed by a progressive increase of HBD-2 (but not HBD-1) up to day 28 [136]. Further studies in preterm neonates demonstrated an impact of feeding pattern on fecal HBD-2 [140].

Cathelicidin is produced by the intestinal epithelial cells only during the fetal and neonatal period, and progressively disappears thereafter [146,170]. The disappearance of LL-37 begins around the 29th week of gestation, coinciding with the progressive increase in Paneth-cell density and the production of HD-5 and HD-6 [72,171].

Gut colonization after birth is an important factor affecting AMP production by the GI epithelium. The balance between “healthy” gut microbiota and enteric pathogens is important for the maintenance of intestinal integrity and maturation of the immune system. Additionally, it inhibits pathogen attachment to gut epithelium and subsequent invasion into the mucosa. The Paneth-cell-derived HD-5 and HD-6 as well as lactoferrin- and lactoferrin-derived AMPs were found to promote a “healthy” gut microbiome. This microbiome includes bifidobacteria and lactobacillus, which in turn upregulate the expression of LL-37, HBD-1, and HBD-2 [46,158,165,169].

In summary, the epithelial and immune cells of the fetal and neonatal GI tract produce a variety of AMPs, namely HD-5 and -6, HBD-1 to -4, LL-37, and antiproteases. Paneth cells are a very important source of α-defensins after the second trimester of pregnancy. Although clinical data on GI-tract AMPs in neonates is sparse, AMPs derived from human milk are presumed to be of utmost importance for neonatal GI antimicrobial protection and immune regulation.

### 3.3. Effect of Labor on AMP Expression

The effect of the delivery mode on AMP levels in AF varies depending on the type of AMP. The expression of HBD-1 in the placenta and chorioamniotic membranes, as well as its levels in AF, have been reported to be unaffected by the process of labor [112,118]. In contrast, HBD-3 and HNP-1 to -3 were found to be significantly higher in AF during labor at term compared to birth without labor [103,131,172]. Similar results have been reported regarding LL-37, which was found to be higher in the umbilical-cord plasma and neutrophils of neonates born vaginally compared with those born by elective caesarean section [144]. In a recent study, analysis at gene and transcript level demonstrated upregulation of AMP gene expression, especially that of elafin, in maternal peripheral blood obtained two days after vaginal delivery compared with caesarean section. [155]. In addition, caesarean section has been correlated with low levels of SLPI in BAF from ventilated preterm infants with respiratory distress syndrome at three and four days of life [154]. With respect to AMPs in breast milk, HBD- 1 and HBD-2 levels in colostrum were significantly higher in women who delivered vaginally compared with caesarean section [173].

In summary, most published data indicate increased synthesis and levels of AMPs during vaginal delivery, which is attributed to the triggering effect of labor stress on the mobilization and release of AMPs. This is probably secondary to the increase in neutrophil count [137,172]. The decreased AMP expression in neonates born by caesarean section may provide an explanation for the adverse immune and inflammatory outcomes reported for mothers and offspring following caesarean section [174,175].

### 3.4. Effect of Prematurity on Fetal and Neonatal AMP Levels

Levels of certain AMPs in the circulation, neutrophils, and tissues have been reported to differ between preterm neonates, term neonates, and adults [109,136,176]. HBD-2 and LL-37 levels in cord blood (reflecting fetal and maternally derived AMPs), neonatal neutrophils, and BAF from ventilated neonates were found to be lower in preterm infants compared to term infants [109,139,143,147]. Decreased levels in preterm infants were also reported for SLPI in BAF [154] and for hepcidin in cord blood and peripheral serum [83]. Expression of HD-5 and HD-6 mRNA in Paneth cells at the 24th week of gestation was approximately 200-fold lower than expressed in term neonates and adults [132]. Moreover, it should be noted that preterm infants born before 28 weeks of gestation are deprived of vernix caseosa, which is rich in AMPs [30]. On the other hand, comparable levels between preterm and term neonates have been reported for HBD-3 in AF (in the absence of intrauterine infection or inflammation); HNP-1, -2, and -3, and SLPI in cord blood and/or neonatal serum; LL-37 in BAF; and HBD-2 in the intestine. HBD-2 levels in feces of preterm infants were found comparable to, or lower than, for term neonates [131,136,137,138,147,149,154,172].

Collectively, published data suggests that levels of certain AMPs in the lungs, GI tract, skin, and circulation of preterm infants are lower than those in term infants and in adults.

## 4. Changes in AMP Expression and Levels in Perinatal Infections and NEC

### 4.1. Changes in AMP Expression and Levels Associated with Chorioamnionitis and Neonatal Infections

Studies in adults and children with sepsis reported significant changes in the expression of AMPs and AMP levels (found in plasma and/or BAF), which supports their involvement in human infections. In adults with sepsis, the endotoxin-inducible HBD-2 mRNA expression in blood cells has been found to decrease while plasma levels increase, indicating that blood cells are not the only source of circulating HBD-2 [177]. Moreover, a recent study in adults with severe sepsis showed decreased mRNA expression and plasma levels of HBD-3, suggestive of a decreased synthesis of this AMP during severe sepsis [178]. In contrast, plasma HNPs -1 to -3 in adults with severe sepsis or septic shock, and in septic children without neutropenia, were significantly increased when compared to controls, obviously reflecting neutrophil accumulation [179,180]. Other studies have associated respiratory infections in adults and children with increased plasma and/or BAF levels of HBDs and LL-37 [181,182,183].

Regarding intrauterine and neonatal infections, several studies have provided evidence for the involvement of α- and β-defensins in chorioamnionitis and neonatal infections (Table 3, Figure 4). Defensin expression in chorioamniotic membranes and defensin levels in AF are induced by inflammatory cytokines (such as IL-1b and TNF-α) and bacterial cell-wall components (such as LPS and peptidoglycans) [101,102,103,104,121,122,184,185]. Early studies using ELISA showed that the levels of HNP-1 to -3 in AF were four-fold to 24-fold higher in the presence of intrauterine inflammation (especially in culture-positive cases), compared with uncomplicated pregnancies [104]. This was confirmed by later studies, in which HNP-1 to -3 levels in AF were found to be significantly increased in preterm prelabor rupture of membranes, microbial invasion in the amniotic cavity, and chorioamnionitis [103]. Additionally, proteomic analysis has provided strong evidence linking increased HNP-1 levels with chorioamnionitis and preterm prelabor membrane rupture [186]. Considering that the HNPs found in AF are produced by neutrophils, 99% of which are of fetal origin, increased levels of HNP-1 and HNP-2 in the AF strongly reflect fetal inflammation, with or without sepsis. Based on these results, HNP-1 and HNP-2 in AF have been investigated as surrogate biomarkers of clinical and subclinical chorioamnionitis, as well as fetal inflammation, even in the absence of infection [111,121,187,188,189,190].

Regarding HBDs, authors have reported increased expression of HBD-1, -2, and -3 in placental and chorioamniotic membranes as well as increased AF levels in response to inflammatory mediators, bacterial components, and intrauterine infection and/or inflammation [112,118,119,121,122,172].

LL-37 levels in AF have also been reported to be significantly increased in pregnancies complicated by chorioamnionitis or other neutrophil-stimulating conditions (e.g., inflammation and labor), when compared with uncomplicated pregnancies. Specifically, early studies demonstrated increased LL-37 levels in AF, reflecting ongoing AF infection and progression to chorioamnionitis [125,191]. Studies which used proteomics have documented that LL-37 is upregulated in the presence of chorioamnionitis and microbial invasion of the AF [186]. The increased LL-37 in AF is derived from the mother and from recruited neutrophils during chorioamnionitis and vaginal delivery. It may provide protection to the fetus and neonate against microbes, such as *S. epidermidis*, which has frequently been isolated from the skin of newborn infants and can cause systemic infections in preterm infants [142,144,145,192]. Other studies showed that LL-37 has a synergistic activity with HBD-2 against group B *Streptococcus*, another important cause of neonatal infection [142,148].

Data are limited regarding the effect of intrauterine infections on antiprotease levels. Three studies reported increased expression of elafin mRNA in the placenta and chorioamniotic membranes, as well as elafin levels in the AF, following exposure to IL-1β and chorioamnionitis [101,118,193]. The importance of SLPI for intrauterine innate defense is supported by studies showing that increased levels of SLPI in vaginal fluid and infant saliva were associated with a lower incidence of maternal–fetal transmission of infectious agents, such as human immunodeficiency virus (HIV) and human papilloma virus (HPV) [194,195]. However, no change was observed in SLPI expression in placenta cells or isolated trophoblasts following either LPS stimulation or incubation with inactivated bacteria in vitro. The authors attributed the lack of response to the absence of TLR4 in the isolated trophoblast [116].

Clinical data on the changes of AMPs in neonatal sepsis and other infections are scarce. Relevant studies in neonates are limited to AMP measurements in the BAF of ventilated preterm neonates. They showed increased levels of HNPs -1 to -4, LL-37, and SLPI in infants with congenital or postnatal pneumonia, and in BAF samples positive for Candida antigen [111,145,152,190]. Only one study examined the cord-blood levels of HBD-2, and found that low levels were associated with increased risk of late-onset sepsis [139]. Moreover, clinical studies demonstrated increased hepcidin-25 serum levels in neonates with sepsis [81,147,156,196].

In summary, studies on pregnancies complicated with chorioamnionitis have documented increased AMP expression and levels in the fetomaternal environment, supporting the involvement of AMPs in intrauterine infection and/or inflammation. The sparse clinical studies involving neonates with sepsis or lung infections demonstrated increased levels of AMPs in BAF or serum.

### 4.2. Changes in Gastrointestinal AMPs Associated with NEC Development and Severity

NEC is the most common cause of GI-related morbidity and mortality in neonates. The disease predominantly affects preterm neonates due to the immaturity of their GI defense system, and preterm birth is among the leading factors predisposing to NEC, along with dysbiosis and prenatal infection/inflammation [197]. In this context, the antibacterial, immunomodulating, anti-inflammatory, and tissue-repairing properties of AMPs could play an important role not only in prevention but also in reducing the severity of the disease.

Experimental studies have shown that defensins and cathelicidin protect the integrity of the intestinal barrier with their tissue-repairing properties, thus regulating intestinal homeostasis [198,199]. Interestingly, NEC tends to develop in preterm neonates at about 26–30 postconceptional weeks, corresponding to the natural shift from LL-37 to defensin production in the fetus [110,146,170,200,201,202]. Around this time, LL-37 synthesis is progressively reduced while the production of α-defensins remains low [132]. Studies in animals confirmed the latter findings and documented that low expression of HD-5 and HD-6 mRNA in Paneth cells was associated with pathological findings similar to those observed in neonatal NEC [165,203]. Studies on intestinal tissues obtained after surgical resection from neonates with NEC and non-NEC controls showed that during NEC the Paneth cells are capable of responding with a three-fold increase of the expression of HD-5 and HD-6 [133]. A more recent study on intestinal tissues from preterm infants with NEC and matched controls demonstrated that in the acute phase of NEC the Paneth cells’ abundance was comparable between the two groups. However, after recovery from NEC, Paneth-cell hyperplasia was observed with concomitant elevated expression of HD-5 mRNA [134]. These findings combined suggest that (a) low production of HD-5, HD-6, and LL-3 in preterm infants predisposes them to NEC, (b) neonatal Paneth cells are capable of responding to infectious stimuli by increasing HD-5 and HD-6 production, and (c) theoretically, exogenous administration of HD-5, HD-6, and LL-37 may have a preventive or even therapeutic effect on NEC. Indeed, the potentially therapeutic effect of cathelicidin on NEC is supported by a recent experimental study, which showed a significant beneficial effect of cathelicidin against intestinal injury [204].

**Figure 4 jcm-11-05074-f004:**
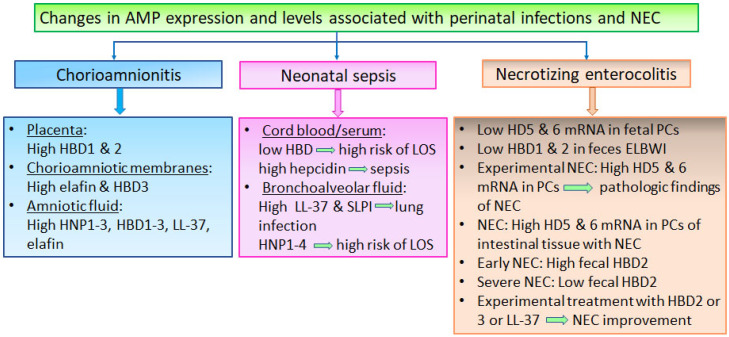
Schematic illustration of published data regarding changes in AMP expression and levels associated with chorioamnionitis, neonatal infections, and necrotizing enterocolitis (NEC). Abbreviations: ELBWI, extremely-low-birth-weight infants; LOS, late onset sepsis; PCs, Paneth cells [81,101,111,112,118,119,120,121,122,123,125,132,133,134,136,139,147,154,156,165,172,184,185,186,191,192,193,196,203].

HBDs are crucial for the gut’s resistance against microbe-triggered GI disease. Studies in extremely-low-birth-weight infants documented low levels of HBD-1 and HBD-2 in the meconium and feces during the first four postnatal weeks [136]. Further analysis in relation to the development and severity of NEC showed that (a) fecal levels of HBD-2 increased significantly 12 to 72 h prior to the clinical signs in patients who developed moderate NEC, suggesting a potential use as an early biomarker of NEC; (b) considerable HBD-2 expression remained for several weeks, probably reflecting an ongoing intestinal immune response to NEC; (c) in patients with severe NEC, HBD-2 levels in feces remained low, at levels similar to those in healthy controls; and (d) HBD-2 protein expression in intestinal samples obtained during surgery was very low during acute severe NEC. The latter two findings indicate that the inability to induce HBD-2 synthesis in response to intestinal infection or inflammation may contribute to the severity of the disease [136]. Low levels of HBD-2 in feces of preterm infants may result in a repression of acquired immunity, eventually leading to impaired T-cell function [67,136,205,206]. In support of this, a recent experimental study demonstrated that the systematic administration of HBD-2 ameliorated intestinal inflammation and alleviated disease symptoms [207]. On this basis, we can speculate that even in extremely-low-birth-weight infants the intestinal epithelium maintains the ability to respond to HBD-2, mitigating the severity of NEC. However, in some infants, a potential dysfunction of the mechanisms regulating HBD-2 synthesis by the intestinal epithelium, such as TLR4 signaling, may lead to impaired HBD-2 production resulting in more severe NEC [116].

Theoretically, HBD-3 could have a beneficial effect on NEC, secondary to its anti-inflammatory properties and its stimulatory effect on enterocyte migration, which helps to preserve the integrity of the intestinal barrier [50,208]. Interestingly, studies using animal models of NEC demonstrated that exogenous HBD-3 reduced the incidence and severity of the disease, and improved outcomes [208,209].

Human milk has been proven to be the only effective means for the prevention of NEC [210]. This can be attributed, in part, to the AMPs contained in human milk that may contribute to NEC prevention and the alleviation of its course, and help with repair of the GI epithelium following injury.

Other important factors associated with the development of NEC are the intestinal microbiota. Changes in the abundance and composition of the gut microbiome have been associated with an increased risk of NEC. In this respect, promotion of healthy intestinal microbiota may decrease the incidence and severity of NEC [211]. It is known that Paneth cells’ α-defensins are directly involved not only in the removal of pathogens but also in establishment of symbiosis with the normal intestinal microbiota [46,169].

Overall, published data supports the involvement of intestinal and milk AMPs in NEC development and outcome. Properties of AMPs that contribute to prevention of NEC and/or mitigation of its severity include their microbicidal and immunomodulatory activity, their bifidogenic effect, and their regenerative and wound healing properties.

Available data regarding the expression AMPs and their levels in tissues and biofluids of the human fetus and neonate with NEC are summarized in Table 3 and Figure 3.

## 5. Clinical Applications of AMPs in Neonates and Future Perspectives

The high mortality and morbidity associated with perinatal infections and NEC, in particular the detrimental effects on the developing brain, make early diagnosis and treatment critical. Efforts to assess AMPs in the intrauterine environment and the fetus or neonate aim to investigate their role in the pathophysiology of pre- and post-natal infection and inflammation, and their potential use in clinical practice as early diagnostic indicators and therapeutic agents. Currently, a considerable number of AMPs are under evaluation as diagnostic and predictive biomarkers, and as adjunctive treatment for sepsis or therapeutic alternatives to antibiotics.

### 5.1. The Potential Role of AMPs as Surrogate Biomarkers of Intrauterine and Neonatal Infection and NEC

Considering the nonspecific clinical signs of neonatal sepsis, reliable biomarkers for early and accurate diagnosis of infections in this vulnerable population are urgently needed [212,213]. Although several biomarkers have been evaluated, no single biomarker known thus far has the desirable characteristics [212,213,214]. Even blood cultures, which are regarded as the gold standard for sepsis diagnosis, have two main disadvantages; the need for 24–72 h to obtain results and the high percentage of false negative results, reported to range between 27% and 92% in neonates with suspected early-onset sepsis [104,188,215,216,217].

In the context of AMP changes in chorioamnionitis, neonatal infections, and NEC, AMP levels in AF and neonatal biofluids including feces could be used as potential biomarkers for the diagnosis and/or prediction of chorioamnionitis and early-onset neonatal sepsis. [136,187,188,215,218].

During the past two decades, advances in proteomics and other omics technologies have opened new horizons in the field of biomarker discovery through simultaneous assessment of multiple proteins and metabolites by using small quantities of plasma or serum [219]. Actually, studies in AF allowed the identification of a set of four AMPs and proteins (HNP-1, HNP-2, and calgranulins S100A8 and S100A12) that could differentiate inflamed from non-inflamed AF. Based on the presence of these peptide and proteins, the authors created a score (mass restricted [MR] score) ranging from zero (no increase of biomarkers) to four (four increased biomarkers). It was found that an MR score of three or four corresponded with high sensitivity and specificity (92.9% and 91.8%, respectively) for detecting intra-amniotic inflammation. In addition, the same group of researchers showed that the number of the detected biomarkers was related to the severity of histological chorioamnionitis, neonatal hematological indices, and the risk of early-onset sepsis [215,218]. These results were confirmed in a more recent study showing that the presence of three or four of these specific biomarkers could identify subclinical chorioamnionitis with an accuracy of 89.7% sensitivity of 81.8%, specificity 100.0%, PPV 100.0%, and NPV 81.0% [187]. A study of 30 neonates with congenital pneumonia showed that an LL-37 cut-off level of 17 pg/mmol could diagnose congenital pneumonia with a sensitivity of 93% and specificity of 86% [145]. The documented diagnostic accuracy of the α-defensins HNP-1, HNP-2, and LL-37 holds promise for their use as novel biomarkers for the detection of subclinical chorioamnionitis. Currently no other AMP in AF has been evaluated for its value as a diagnostic or predictive biomarker of chorioamnionitis and early-onset sepsis.

Although clinical studies in neonates associated sepsis and/or NEC with levels of HBD-1, HBD-2, and hepcidin in cord blood, serum, or feces [81,136,139,156,196], the diagnostic accuracy of these AMPs in such situations has not been evaluated.

In conclusion, existing though limited data indicate a potential utility of certain AMPs, including HNP-1 and -3, as predictive or diagnostic biomarkers in chorioamnionitis. However, the diagnostic value of AMPs in neonatal sepsis and NEC has not been fully evaluated. The scarcity of relevant clinical studies performed so far can be attributed not only to ethical reasons related to blood sampling from neonates, but also to insufficient knowledge on the pathophysiology of neonatal sepsis and NEC. Intensive and in-depth research using proteomics, metabolomics, genomics, and transcriptomics at different stages of sepsis and/or NEC could potentially contribute to further elucidation of the underlying pathophysiologic pathways of sepsis and NEC, leading to the discovery of sets of AMPs, perhaps in combination with other biomarkers, that could accurately predict and facilitate early diagnosis of chorioamnionitis and neonatal sepsis and NEC.

### 5.2. The Role of AMPs as Therapeutic Agents in Neonatal Sepsis and NEC

In the context of increasing antimicrobial resistance of pathogens causing infections in the neonatal intensive care setting, the discovery of effective new antimicrobials is challenging. The reported effectiveness of AMPs against drug-resistant microbes, along with the association of low AMP levels in neonates with severe NEC, highlight the potential role of AMPs as candidates for new antimicrobial drugs. They could be extremely useful in high-risk neonates exposed to antibiotic-resistant pathogens within neonatal intensive care units [4,130,136,146,170,220,221].

Studies using experimental models of sepsis showed that administration of LL-37 improved survival in a dose-dependent manner by inhibiting bacterial growth, regulating production of inflammatory cytokines, and playing a role in the activation and pyroptosis of macrophages [53,222]. Additional studies in a gut-injury model showed that administration of cathelicidin significantly improved intestinal barrier function, preserved survival of intestinal stem cells known to participate in epithelium regeneration after intestinal injury, prevented proliferation arrest, promoted the growth of isolated intestinal crypts. These findings indicate a potentially preventive and therapeutic role for LL-37 in NEC [43,204,223]. In vitro studies in human cord blood showed that, compared with term neonates, the blood in preterm neonates had impaired endogenous killing capacity against *S. aureus* and *S. epidermidis*, which was strongly enhanced after the addition of LL-37 [114]. Likewise, the administration of either HBD-2 or HBD-3 to experimental NEC models ameliorated intestinal inflammation and improved not only the incidence and severity of NEC but also the survival rate [207,208]. Another experimental study indicated that pretreatment with HBD-3 could significantly inhibit enterocyte autophagy and promote epithelial cell migration [209].

Nevertheless, clinical studies have thus far evaluated only the potential effect of oral lactoferrin (partly reflecting the effect of lactoferrin-derived AMPs) on neonatal sepsis, NEC, and mortality [224]. The most recent systematic review of this issue, including the results of the ELFIN study, concluded that lactoferrin supplementation significantly reduced the incidence of late-onset sepsis, but had no significant effect on all-cause mortality and neurodevelopmental outcomes at 24 months of age in preterm infants [224,225].

An additional therapeutic potential of AMPs in neonates relates to the effects of the antiproteases elafin and SLPI on oxygen- and ventilator-induced lung injury and the development of bronchopulmonary dysplasia. Lung infection, inflammation, and elastase activity are important contributors to the development of bronchopulmonary dysplasia. In this respect, theoretically, the administration of either elafin or SLPI to mechanically ventilated neonates could prevent and/or alleviate lung injury, due to the substances’ anti-elastase, anti-infective, and inflammation-regulating properties. Experimental and clinical studies in patients with cystic fibrosis have shown that recombinant SLPI, administered via aerosolization, retained its structural integrity and antiprotease function and can reduce neutrophil-elastase-related damage to the respiratory epithelium [226]. Likewise, experimental studies in newborn mice undergoing mechanical ventilation and hyperoxia showed that elafin treatment inhibited alveolar cell apoptosis and promoted alveologenesis, thus enabling lung growth and decreasing the risk of chronic lung disease [227,228]. It remains to be examined whether elafin and SLPI have a therapeutic and preventive effect onventilator-induced lung damage in high risk preterm infants.

The use of exogenous AMPs for the prevention and treatment of neonatal infection- and inflammation-associated morbidities is challenging. Modifying natural AMPs or creating new peptides is not an easy task, and multiple barriers are to be overcome. Modification of the AMPs’ structures aims to increase their resistance to proteolytic enzymes in vivo, enhance their antimicrobial spectrum and effectiveness against multi-drug resistant pathogens, and decrease toxicity. Additional issues to be resolved include AMP kinetics and suitable administration methods. The high manufacturing cost and the method of purification are also important concerns [6,220,229,230,231,232]. Currently, only a few AMPs have been approved by the United States Food and Drug Administration (FDA), including glucopeptide antibiotics (vancomycin, teicoplanin, oritavancin, dalbavancin, and telavancin), daptomycin, polymyxins (polymyxin B and colistin), and gramicidin for use against gram-positive and gram-negative antibiotic-resistant bacteria. Additionally, nikkomycins and echinocandins are under investigation as antifungal drugs [8]. Moreover, several synthetic and natural AMPs are currently under evaluation in pre-clinical and clinical studies, the analysis of which is beyond the scope of this review [6,233].

Regarding the clinical application of new AMP-based antimicrobials in neonates, in addition to the technical difficulties mentioned above, the bi-directional effect of certain AMPs, for example β-defensins and LL-37 [55,62,234], on inflammatory and immune responses may be a source of additional concern. Considering the immature immune and inflammatory responses of preterm neonates, a balance between the pro- and anti-inflammatory activity of exogenous AMPs may be difficult to achieve [116]. These limitations and concerns may have contributed to the lack of clinical studies on the use of exogenous natural or synthetic AMPs for the prevention and/or treatment of neonatal sepsis and NEC [109,110,130,136,220,221].

Overall, clinical studies regarding the potential use of either natural or synthetic AMPs for the treatment of neonatal sepsis and NEC are sparse or lacking. Therefore, no AMP analogs have been approved by the FDA for use in neonates. However, in some neonatal intensive care units, certain synthetic AMPs, such as vancomycin, teicoplanin, colistin, and echinocandins, are being used off-label in neonates with sepsis caused by multi-drug resistant microbes and fungi. An alternative to AMP administration may be the induction of their expression by specific agents such as vitamin D, L-isoleucine, and butyrate [143]. Existing data are controversial, probably due to the rather weak effect of these agents on the expression of AMPs [235]. Other new technologies, such as nanotechnology, could help pharmaceutical companies to improve the antimicrobial effect, immunomodulating properties, tolerance, and delivery of new AMP-based antibiotics, targeting effective and safe use in neonates [223,232,236,237,238,239,240]. For the time being, these novel approaches remain untested in neonates [7,10].

## 6. Conclusions

Existing data indicate that AMPs are produced by several components of the female reproductive system during pregnancy, as well as by the fetus and neonate. AMPs constitute an important part of the intrauterine and neonatal innate immune system. However, fetal and neonatal AMP levels, especially in preterm infants, are lower than in adults, predisposing them to infections and NEC. Reported changes in AMPs during chorioamnionitis, neonatal sepsis, and NEC strongly suggest their involvement in the pathogenesis of prenatal and neonatal infections and NEC, and also their potential utility as predictive and diagnostic biomarkers and as preventive and/or therapeutic agents. However, clinical investigations in pregnancy and neonates have been minimal, probably due to important ethical and practical considerations. Future studies should include the use of modern technology, such as omics and nanotechnology, to overcome these limitations and further elucidate AMPs’ role in the pathogenesis and diagnosis of perinatal infections and NEC. Hopefully, this will lead to the production of effective and safe synthetic AMP analogs that may be used as therapeutic or preventive agents. Particularly in light of multi-drug resistant pathogens, the availability of natural AMP analogs with improved antibacterial and physicochemical properties and decreased toxicity could be vital in the treatment and/or prevention of neonatal sepsis and NEC, two devastating morbidities in preterm infants.

## Figures and Tables

**Figure 1 jcm-11-05074-f001:**
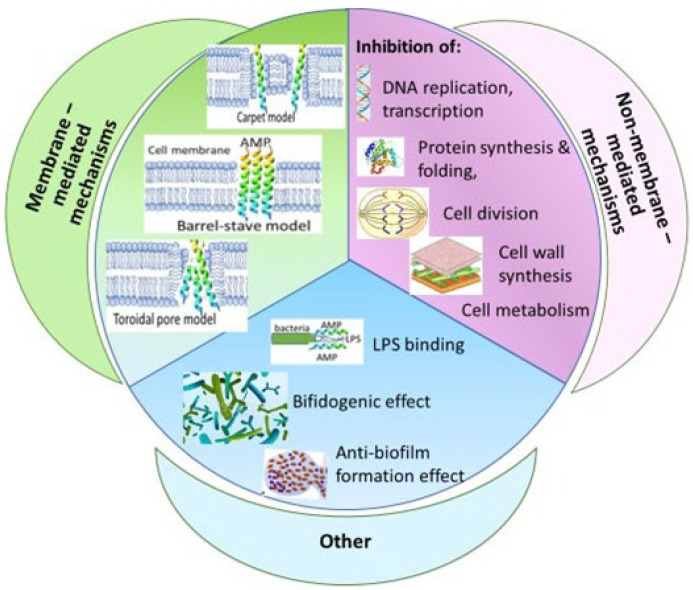
Mechanisms of antimicrobial actions of human antimicrobial peptides.

**Figure 2 jcm-11-05074-f002:**
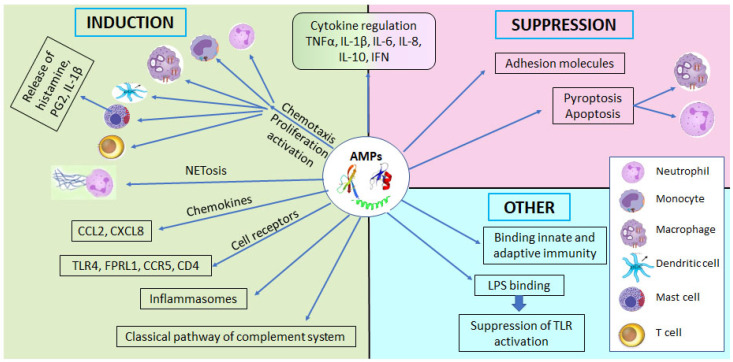
Summary of immunomodulating properties of antimicrobial peptides.

**Figure 3 jcm-11-05074-f003:**
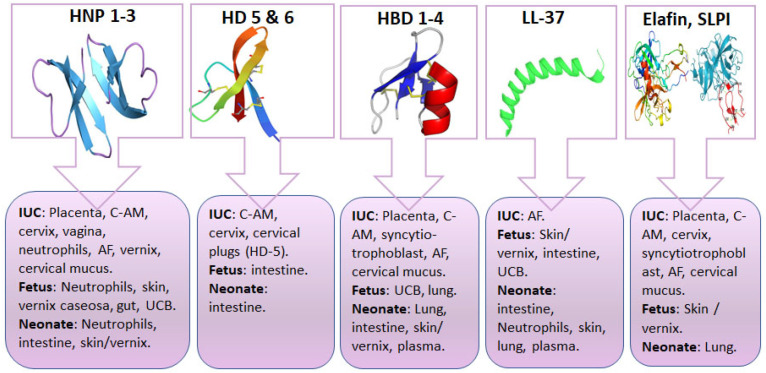
Sites of antimicrobial peptide detection in the maternal reproductive system during pregnancy, and in the fetus/neonate. Abbreviations: AF, amniotic fluid; C-AM, chorio-amniotic membranes; HBDs, human β defensins; HD, human defensin; HNP, human neutrophil peptide; IUC, intrauterine components; SLPI, secretory leukocyte peptidase inhibitor; UCB, umbilical cord blood.

**Table 1 jcm-11-05074-t001:** Sources and properties of the main human antimicrobial peptides (AMPs).

AMPs	Sources	Antimicrobial Spectrum	Immunomodulating Actions	Other Properties
Human neutrophil peptides (HNPs) 1 to 4	Neutrophils	Gram+ and gram- bacteria (*S. aureus, bacillus subtilis, S. epidermis, E. coli, P. aeruginosa*), fungi (*C. albicans*), and viruses (*influenza v., HSV, CMV*)	Induction: TNF-α and IL-1β and chemotaxis (neutrophils, immature Dcs and other immune cells). Inhibition: IL-10.	
Human defensins (HDs) 5 & 6	PCs (HD-5 and -6), epithelia of the female reproductive tract (HD-5)	Gram+ and gram- bacteria *(E. coli, Listeria monocytogenes, Salmonella typhimurium, S. aureus)*, and *C. albicans).*	Induction: IL-8 and chemotaxis of macrophages, T lymphocytes, and mast cells.	Modulation of the commensal bacteria in the small intestine.
Human β-defensins (HBDs) 1 to 4	Neutrophils and other immune cells, keratinocytes, and epithelia of respiratory, GI, and genitourinary tracts. They can be found in blood, urine, heart, and skeletal muscles (HBD-3), and testis.	HBD-1 to -4: Gram- bacteria (*P. aeruginosa, E. coli, vancomycin resistant Enterococcus*); HBD-1: anaerobic gram+ bacteria. HBD-2 to -4: Gram+ bacteria (S. aureus, S. Mutans, *Str. Pneumoniae,* Str. pyogenes)	Induction: pro-inflammatory cytokines, chemotaxis of inflammatory cells, differentiation of monocytes, proliferation and activation of CD4+ T cells, activation of mast cells (release of histamine & PGD2). Inhibition: IL-6 and IL-8 (HBD-3), apoptosis of DCs. Linkage of innate with acquired immunity, activation of the classical complement system pathway.	Preservation of epithelial barrier integrity, amelioration and repair of inflammation-induced tissue injury, antioxidant action.
Cathelicidin LL-37	Immune cells (neutrophils, macrophages, monocytes, B-cells, T-cells), and in most types of epithelial cells (GI, skin, lung, etc.).	Bacteria (*E. coli, Listeria monocytogenes, Enterococcus faecium)*, fungi, and viruses. Synergistic effect with HNP-1, HBD-2, and HBD-3. Inhibition of biofilm formation.	Induction: Production of IL-1β, IL-6, IL-8, and TNF-α, IL-10, and chemokines, chemotaxis of neutrophils, monocytes, and mast cells, monocyte differentiation, macrophage pyroptosis and activation, vascular endothelium proliferation. Suppression: neutrophil apoptosis stimulates bactericidal activity. Inhibits sepsis-induced production of pro-inflammatory cytokines. Binds LPS (antiendotoxin action).	Promotion of angiogenesis, arteriogenesis, and re-epithelialization of wounded epithelia and epidermis.
Antileukoprotease elafin	Epithelia (skin, respiratory tract, intestine, endometrium), neutrophils, and macrophages.	*S. Aureus, P. aeruginosa, A* *spergillus fumigatus, and C. albicans.*	Promotion of neutrophil and lymphocyte chemotaxis, LPS response, humoral and cellular aspects of adaptive immunity. Inhibition of inflammatory cell recruitment and NF-*κ*B activation.	Inhibition of proteases, promotion of tissue remodeling and cellular differentiation.
Antileukoprotease SLPI	Inflammatory cells (neutrophils and macrophages, mast cells), keratinocytes, and epithelial cells of respiratory and GI systems, and amniotic membranes.	Gram+ bacteria (*S. aureus and S. epidermidis, group A Streptococcus*), Gram- bacteria (*E. coli**, P. aeruginosa*), fungi (*Aspergillus fumigatus,* *C. albicans*), and viruses.	Inhibition of inflammatory infiltrate, NF-*κ*B activation, mast cell histamine release, and C5a production. Modulation of adaptive immune responses.	Neutralization of proteases, involvement in cutaneous and oral mucosal wound healing.

C, Candida; CB, cord blood; DCs, dendritic cells; E, Escherichia; GI, gastrointestinal; P, Pseudomonas; PCs, Paneth cells; S, Staphylococcus; SLPI, secretory leukoprotease inhibiting peptide.

**Table 2 jcm-11-05074-t002:** Antimicrobial peptides derived from human milk proteins.

HM Protein	Production of HM Protein-Derived AMPs	Antimicrobial Spectrum	Ref.
a-lactalbumin (La)	Digestion of La with trypsin releases 2 AMPs: α-La f(1–5)/LTD1, and α-La f(17–31) S-S(109–114)/LTD2. Digestion with chymotrypsin releases AMP α-La f(61–68)S-S(75–80)/LCD.	Mainly gram+ bacteria (*Bacillus subtilis, S. epidermidis, S. lentus*); α-La f(17–31) S-S(109–114)/LTD2 and α-La f(61–68)S-S(75–80) were also effective against *S. aureus, K. pneumoniae, and P. aeruginosa.*	[86,87,88]
Casein	Digestion of casein with chymocin, proteolysis, or acidification releases the AMPs casecidin, lactenin, isracidin, caseicin A and B, kapacin, and κ-casecidin. Fermentation by *L. acidophilus* was found to produce casein A, B, and C.	Gram+ and gram- pathogens (*S. species, Str. pneumoniae, Listeria, Str. pyogenes, E. coli, Enterobacter faecalis, and P. aeruginosa*).	[87,90,91]
Lactoferrin	Digestion of lactoferrin with pepsin produces lactoferricin (amino acids 1–40 of lactoferrin) and lactoferricin-derived shorter AMPs.	Effective against gram+ and gram- bacteria (S. aureus, Str. Mutans, *E. coli*), and viruses (HSV-1 & -2, CMV, HPV).	[87,89]
	Human lactoferrampin: Synthetic peptide with an amino-acid sequence corresponding to 269–285 amino acids of human lactoferrin.	Broad spectrum of antibacterial activity, although some bacteria are resistant to this peptide (*E. coli and Str. sanguis*).	[17,89]

CMV, cytomegalovirus; E, Escherichia; HPV, human papilloma virus; HSV, herpes simplex virus; K, Klebsiella; P, Pseudomonas; S, Staphylococcus; Str, Streptococcus.

**Table 3 jcm-11-05074-t003:** Studies regarding AMP expression and levels in human fetuses and neonates in health, sepsis, and NEC.

AMPs	First Author & Year [Ref.]	Aim	Study Design and Population/Material	Main Results	Reference
HNPs 1–4 in BAF	Tirone C, et al., 2010	HNP -1 to -4 in BAF and ventilator-associated pneumonia.	Cohort study of 24 PTI (GA <30 wks), nine with pneumonia. Proteomics.	HNP-1 and -2 were detectable in all samples, and were increased in the pneumonia group.	[111]
HNP-1 to -3	Faust K, et al., 2014	Expression of HNP-1 to -3 in CB, and influencing factors.	Cohort study of 139 preterm (GA 24–36 wks) and 36 term infants (*n* = 36). HNP-1 to -3 in supernatants of whole-CB cultures.	Increased CB HNP-1 to -3 in clinical chorioamnionitis.	[131]
HD-5 & HD-6	Mallow EB, et al. 1996	HD-5 & -6 mRNA expression in PCs of the fetus.	Intestinal tissue from fetuses with GA 19–24 wks.	HD-5 and -6 mRNA detected in fetal PCs cells from 13.5 wks of GA, 40- to 250-fold less than in adults.	[132]
HD-5 and HD-6	Salzman NH, et al. 1998	HD-5 and HD-6 expression in PCs of NEC cases and controls.	Case-control study. Six NEC-cases (GA 25–31 wks) and five controls (GA 35–40 wks).	HD-5 expressed at 24 wks of GA at levels lower than term infants and adults, and increased three-fold in NEC cases.	[133]
HC-5 and PCs	Puiman PJ, et al., 2011	PC developmental changes in PTI with NEC	Intestinal tissue from 55 PTI with NEC, 22 preterm controls, and nine term controls.	Acute NEC, no effect. After NEC recovery, PC hyperplasia and elevated HD-5 expression.	[134]
HC-5 and PCs	Heida FH, et al., 2016	Developmental changes in PC and HD-5 expression.	Studied 57 samples of ileum tissue from fetuses/infants (GA 9–40 wks).	PCs expressing HD-5 observed at GA >29 wks.	[135]
HBD-1	Wu J, et al., 2019	Immunoregulatory function of HBD-1 in NCBM-dDC&TC.	In vitro; NCBM-dDC&TC from human CB.	HBD-1 promotes the differentiation and maturation of DCs, inhibits apoptosis of CBM-dDC, promotes proliferation and activation of CB CD4 + T cells.	[78]
HBD-1 and HBD-2	Jenke ACW, et al., 2012	Expression of HBD-1 and -2, IL-8, and TLR4 in NEC.	Cohort study of 68 ELBW infants (GA <27 wks); 12 with NEC, 56 without. Serial stool samples, and intestinal biopsies.	Fecal HBD-1 levels were low in all neonates, HBD-2 levels were increased in chorioamnionitis and moderate NEC (before clinical symptoms) but low in severe NEC.	[136]
HBD-2	Richter M, et al., 2010	Developmental changes in HBD-2 levels in stool from neonates.	Case-control study of 59 preterm and term infants. Stool samples collected between days three and 28.	HBD-2 levels increased significantly between 24 and 42 wks of GA and were not affected by sex or mode of delivery.	[137]
HBD-2	Campeotto F, et al., 2010	Levels of HBD-2 in feces of term and preterm infants and effect of intestinal distress.	Case–control study of 30 healthy term and 20 PTI. Fecal samples (up to day 30 or 60). Case-control study of 10 PTI with intestinal distress and 20 controls.	Fecal HBD-2 did not differ either between healthy term and preterm infants or between infants with clinical intestinal distress and controls, although it was increased in two out of three infants with NEC, and on out of seven with rectal bleeding.	[138]
HBD-2	Olbrich P, et al., 2013	HBD-2 levels in CB and its impact on sepsis.	Cohort study; 42 term and 31 preterm neonates.	HBD-2 was lower in preterm than in term infants. Low HBD-2 was associated with neonatal sepsis.	[139]
HBD-2 and gut microbiota	Corebima BIRV, et al., 2019	Fecal HBD-2 and gut microbiota in PTI in relation to feeding patterns.	Cross-sectional study of 44 PTI, four groups related to type of milk feeding.	The formula milk group had the highest HBD-2, not correlated with microbiota.	[140]
HBD-3	Bian T, et al., 2017	Effects of HBD-3 on HUVECs triggered by TNF-*α* and inflammatory response.	In vitro. HUVECs culture.	HBD3 reduced production of inflammatory mediators and ROS by HUVECs, and inhibited NF-*к*B activation.	[50]
LL-37	Braff et al., 2005	Effects of LL-37 on neonatal human keratinocytes.	In vitro study. Gene expression in keratinocytes after exposure to LL-37.	LL-37 affected the expression and release by keratinocytes of several chemokines and cytokines.	[141]
LL37	Nelson A, et al., 2009	Effects of LL-37 on growth of *S. epidermidis*.	Skin swabs for cultures from 21 term neonates (12 with erythema toxicum).	LL37 was constitutively expressed in the skin, and significantly inhibited growth of *S. epidermidis*.	[142]
LL-37	Misawa Y, et al., 2009	LL-37 expression in neutrophils and plasma levels, and effect of 1a(OH)D3.	Included 25 neonates, 25 adults, and CB, as well as human myeloid leukemia cell line.	Expression of LL-37 was impaired in neonates, and was induced by addition of 1a(OH)D3.	[143]
LL-37	Mandic- Havelka A, et al., 2010	LL-37 levels in CB neutrophils and maternal and neonatal plasma; relation with delivery mode and biochemical markers.	Cohort study of 115 term infants (47 with elective CS) including 50 mother–infant pairs.	In vaginal delivery, cord plasma LL-37 was higher than in CS and was similar to maternal levels. In CS, cord LL-37 was lower than maternal levels. Cord LL-37 was correlated to plasma levels.	[144]
LL-37 & 24(OH)D	Gad GI, et al., 2015	Diagnostic value of LL-37 in congenital pneumonia and in relation to (25 OH)D.	Case-control study; 30 neonates with pneumonia and 30 controls. Serum LL-37 and 25(OH)D assessed.	In congenital pneumonia, LL-37 increased and 25(OH)D decreased. Diagnostic value of LL-37 (cut-off level 17 pg/mmol): 93% sensitivity, 86% specificity.	[145]
LL-37	Scheid A. et al., 2018	Effects of LL-37 on antimicrobial activity in human newborn CB.	Cross-sectional study. 30 neonates (22 term, eight preterm). Antimicrobial activity tested before and after addition of LL-37.	Preterm CB had impaired antibacterial capacity against *S. aureus, S. epidermidis,* and *Candida Albicans,* which was enhanced by LL-37.	[114]
LL-37 and HNP-1–3	Kai-Larsen Y, et al., 2007	LL-37 and HNP-1 to -3 levels and antimicrobial activity of meconium vs neonatal feces.	Cross-sectional of 20 healthy breast-fed term neonates.	Meconium exhibited higher antimicrobial activity against *E. coli* and *GBS* than did neonatal feces. LL-37, HNP-1–2, and HD-5 were present in both meconium and feces; LL-37 higher in feces than in meconium.	[146]
LL-37 and HNP-1 to -3	Strunk T, et al., 2009	LL-37 and HNP-1 to -3 in CB and maternal blood, and their relation with GA and sepsis.	Cohort study of 105 neonates and 100 mothers.	LL-37 in PTI was lower than in term and maternal plasma. HNP-1 to -3 in neonates were lower than maternal levels. AMP levels were not correlated with chorioamnionitis or delivery mode.	[147]
LL-37 and HBD-1	Marchini G, et al., 2002	LL-37 and HBD-1 in skin and vernix caseosa of neonates with erythema toxicum.	Cross-sectional study. Skin biopsies of four term neonates with erythema toxicum and four controls, and vernix caseosa of six healthy infants.	LL-37 (inducible) and HBD-1 (constitutive) were expressed in dermal layer cells in erythema toxicum biopsies. LL-37 was detected in the vernix caseosa, also exhibiting antibacterial activity.	[148]
HBD-1, HBD-2, and LL-37	Schaller-Bals S, et al., 2002	HBD-1, HBD-2, and LL-37 in tracheal aspirates of term and preterm newborns with and without respiratory infections.	Cohort study of 45 ventilated newborns (GA 22–40 wks). Serial BAF samples were obtained daily during mechanical ventilation.	LL-37, HBD-1, and HBD-2 were detected in BAF. Their levels were comparable between term and preterm newborns, correlated with each other and with IL-8 & INF-α, and were increased in pulmonary or systemic infections.	[149]
Cathelicidin (CRAMP) and β-defensins	Dorschner RA, et al., 2003	Expression of cathelicidin and β-defensins in skin of mice and human neonates.	In vitro study in skin of embryonic and newborn mice, and human newborn foreskin.	Cathelicidin expression was increased in the perinatal period. HBD-2 was present in newborn skin. LL-37 and HBD-2 had synergistic activity against *GBS*.	[150]
HBDs and LL-37	Starner TD, et al., 2005	Development and antimicrobial spectrum of HBD-1, -2, and -3, and LL-37, in the neonatal lung.	Midgestational fetal lung explants (GA 18–22 wks), and tracheal aspirates at birth, seven months, and 13 years of age.	HBD-2 and HBD-1 expression was detected in term and postnatal tissues, but not in prenatal tissues. HBD-3 was not detected. LL-37 was expressed in tissues from all developmental ages.	[109]
HBD-1, HBD-3, and LL-37.	Gschwandtner M, et al., 2014	Expression and regulation of HBDs and LL-37 in fetal, neonatal, and adult keratinocytes.	In vitro study in cultured keratinocytes from fetal skin (GA 20–23 wks), neonatal foreskin, and adult skin.	The expression of HBD-2, HBD-3, and LL-37 was significantly higher in keratinocytes from fetal skin than in postnatal skin, and further increased after stimulation.	[151]
HBD-1, HBD-2 and LL-37	Strunk T, et al., 2017	HBD-1, HBD-2, and LL-37 plasma levels in PTI, and effects of *Bifidobacterium breve* supplementation.	Cohort study of PTI (GA <30 wks). Plasma on days 1, 14, 28, and stool prior to and 21 days after probiotic supplementation.	Stool, plasma, and stimulated blood AMP levels changed significantly during the first month of life. Probiotic supplementation did not affect AMP levels.	[110]
SLPI in BAF	Ohlsson K, et al., 1992	The importance of SLPI in protecting against ventilator-induced lung damage in neonates.	Cohort study. 38 ventilated neonates (25 RDS without BPD, 10 RDS and BPD, three pneumonia). Serial SLPI measurements in BAF.	Infants with pneumonia had higher levels of SLPI and elastase in BAF than those with RDS. Infants who also developed BPD had intermediate values.	[152]
SLPI and BPD	Watterberg KL, et al., 1994	Relation of SLPI with neutrophil counts, elastase activity, and BPD in neonates.	Prospective cohort study of 41 neonates; 24/41 developed BPD. Serial BAF samples up to day 28.	During the first week of life, SLPI levels were similar between BPD and no-BPD groups. Neutrophil counts and elastase activity were higher in the BPD group.	[153]
SLPI	Sveger T. et al., 2002	Effect of protease/protease inhibitor balance and neutrophil activity in BAF on RDS and BPD.	Cohort study of ventilated PTI with RDS (*n* = 43). BAF obtained in the first wk of life. Elastase and SLPI were determined.	BPD correlated with low SLPI (*p* = 0.03) (and other anti-protease levels) at seven or eight days of age. CS was correlated with low levels of SLPI on days three and four.	[154]
Elafin, HNP-1 to -4	Kothiyal P, et al., 2020	Delivery mode relation to postpartum expression of elafin and HNP-1 to -4 genes.	Cross-sectional study of 324 mothers delivering at term; 181 had vaginal delivery and 143 CS.	AMPs were upregulated in vaginal delivery compared to CS either with or without labor.	[155]
Pro-hepcidin	Yapakci E, et al., 2009	Relation of serum pro-hepcidin in septic neonates with serum iron parameters.	Case–control study of 15 septic PTI, 17 healthy PTI, six septic term and 16 healthy term neonates.	Pro-hepcidin was significantly increased in preterm and term neonates with sepsis. Pro-hepcidin was not correlated with iron parameters.	[81]
Hepcidin	Cizmeci MN, et al., 2014	The value of CB hepcidin levels as a biomarker for early-onset neonatal sepsis.	Pilot data of a prospective cohort study. 38 septic infants (GA 24–41 wks), 20 term and 18 preterm.	Newborns with early onset sepsis had an increased range of CB hepcidin, that was not correlated with GA or markers of anemia.	[156]
Pro-hepcidin	Celik HT, et al., 2015	Association of serum pro-hepcidin in PTI with oxygen radical diseases (i.e., BPD, ROP, NEC).	Case–control study. 80 PTI (GA 25–34 wks); 38/80 with oxygen radical diseases and 42/80 controls.	Pro-hepcidin levels were increased in neonates with BPD or ROP, but not NEC. They were lower in preterm than in term newborns, and not correlated with iron parameters.	[83]
HNP1–3, HBD-2, l LL37, SLPI, lactoferrin	Akinbi HT, 2004	HNP-1 to -3, HBD-2, LL37, SLPI, lactoferrin, and lysozyme in vernix caseosa and amniotic fluid in the absence of chorioamnionitis.	Cohort study of term infants delivered via elective CS without labor or PROM. 25 samples of vernix and 10 of amniotic fluid were collected.	HNP-1 to -3, SLPI, lactoferrin, and lysozyme were identified in vernix suspensions and amniotic fluid; HBD-2 and LL-37 were not detected.	[106]

Abbreviations: BAF, bronchoalveolar fluid; BPD, bronchopulmonary dysplasia; CB, cord blood; CS, caesarean section; DCs, dendritic cells; ELBW, extremely low birthweight; GA, gestational age; HBD, human β defensin; HD, human defensin; HNP, human neutrophil peptide; HUVECs, human umbilical vein endothelial cells; la, lactoferrin; NCBM-dDC&TC, neonatal cord-blood monocyte-derived dendritic cells and T Ccells; NEC, necrotizing enterocolitis; PCs, Paneth cells; PTI, preterm infants; RDS, respiratory distress syndrome; ROP, retinopathy of prematurity; ROS, reactive oxygen species; SLPI, secretory leukocyte protease inhibitor; wks, weeks.

## Data Availability

Data sharing is not applicable to this article. All data included and analyzed in this study have been published and can be found in the cited references.

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
