# Peer review of "Antimicrobial Peptides in Early-Life Host Defense, Perinatal Infections, and Necrotizing Enterocolitis—An Update"

_jcm, 2022, doi:10.3390/jcm11175074_

Round 1

Reviewer 1 Report

1.       This review is too long, however, the study area is interesting. The authors suggested reducing the length of the review by avoiding the repetition of the same information again and again. Authors need to restructure the manuscript more logically.

2.       Authors should remove all repetition such as Immunomodulatory properties for all AMPs can be described in a single section including one illustration. Overall, this manuscript needs a huge reformatting or presentation of data before publication.

3.       Mother and baby both are expressing different sets of AMPs during pregnancy. This is presented all together in the present manuscript. Authors should arrange the manuscript in such a way that this should be present. Also, authors can add a table or an illustration to show the different AMPs expressions of mother and baby during the pregnancy.

4.       Abstract …defensins α and β,….it should be written as α and β defensins.

5.       To make the review concise, the author has the opportunity to include a summarized table, such as 1) AMPs expressed by different organs during pregnancy. 2) AMPs and associated neonatal disease and so on. Adding the information in a tabular format and concise way improves the readability of the review and significantly reduces the length too. Please revise.

6.       Details provided in Table 2 are just repetitions of the information, it should be summarized in the text or just presented in a concise way rather than putting detailed sentences. Please update.

7.       Reference should move to the last column in table2 and other incorporated tables too.

8.       This is a review article and the presentation of a lot of information in the illustration is a must to make it more interactive. For example, 1) different kinds of AMPs, authors can use the structures of available human AMPs. 2) Immunomodulatory mechanisms of AMPs. Provide figure 1 is less informative, please update, there are a lot of examples available. 3) Role of AMPs in neonatal disease, this is important, there is an opportunity to include a nice illustration.

9.       Section 3.3 is interesting, however, the title is too long please rewrite it in a focused and concise way.  

10.   ….human immunodeficiency virus….should it be written in italics? Please confirm and update.

11.   Pseudomonas (P) aeruginosa… (P) should be removed and Streptococcus [Str.] pneumoniae…. Please check the whole manuscript for typographical mistakes.

12.   Number of references is too much. Please reduce. 

Author Response

Dear reviewer,

thank you for the time you spent to review our manuscript and for your comments that helped us to improve the presentation of our manuscript. Please find following our responses to your comments:

  1. This review is too long, however, the study area is interesting. The authors suggested reducing the length of the review by avoiding the repetition of the same information again and again. Authors need to restructure the manuscript more logically”.

RESPONSE. The manuscript has been extensively revised according to the reviewers’ suggestions. We have shortened (by 17%) and restructured our presentation. The length of the revised paper if about 8300 words, a number that may be rather high, but not too high for a review article. In addition, the number of references has been decreased.

  1. Authors should remove all repetition such as Immunomodulatory properties for all AMPs can be described in a single section including one illustration. Overall, this manuscript needs a huge reformatting or presentation of data before publication”.

RESPONSE. The description of immunomodulating properties of defensins and LL-37 is now shorted and it is presented in a single section. However, the antileukoproteases and hepcidin have certain unique actions, that are described separately. Moreover, a table (Table 1) summarizing the AMP properties and an illustration focusing on the immunomodulating properties of AMPs (Figure 2) have been added in the main text.

  1. Mother and baby both are expressing different sets of AMPs during pregnancy. This is presented all together in the present manuscript. Authors should arrange the manuscript in such a way that this should be present. Also, authors can add a table or an illustration to show the different AMPs expressions of mother and baby during the pregnancy”.

RESPONSE: The AMPs produced by the various compartments of the maternal reproductive system during pregnancy and those produced by the fetus/neonate are actually described in different sections of the review, i.e sections 3.1 and 3.2, respectively. In section 3.2, the AMPs produced by the fetus are presented together with those produced by the neonate, since the neonatal life is the natural continuation of fetal life. Moreover, an illustration has been added (Figure 3), in which the AMPs originating from the maternal reproductive system during pregnancy, the fetus, and the neonate are reported separately.

  1. Abstract …defensins α and β,….it should be written as α and β defensins

. RESPONSE: this has been changed.

  1. (a)“To make the review concise, the author has the opportunity to include a summarized table, such as 1) AMPs expressed by different organs during pregnancy”. RESPONSE: The AMPs expressed by different parts of the female reproductive system are shown in Fig. 3. (b)” AMPs and associated neonatal disease and so on.” RESPONSE: All clinical studies regarding the AMPs of fetus and neonate as well as studies on neonatal tissues and in vitro studies on isolated neonatal cells, with or without infection/NEC, are summarized in Table 3 (former Table 2). Presentation of existing data in Table 3 has been considerably shorten and it is now more concise. Moreover, an illustration (Figure 4) summarizing the changes of AMPs during chorioamnionitis, neonatal infections, and NEC has been added.
  2. Details provided in Table 2 are just repetitions of the information, it should be summarized in the text or just presented in a concise way rather than putting detailed sentences. Please update”.

RESPONSE: To our opinion, table 3 (the revised former table 2) is very important since it provides the reader with a quick overview of all published clinical data regarding the AMPs expression in the fetus and neonate thus far. Therefore, we revised Table 3 making it more concise and considerably shorter.

  1. Reference should move to the last column in table2 and other incorporated tables too”.

RESPONSE: references in Table 3 (former table 2) have been moved to the last column.

  1. This is a review article and the presentation of a lot of information in the illustration is a must to make it more interactive. For example, 1) different kinds of AMPs, authors can use the structures of available human AMPs. 2) Immunomodulatory mechanisms of AMPs. Provide figure 1 is less informative, please update, there are a lot of examples available. 3) Role of AMPs in neonatal disease, this is important, there is an opportunity to include a nice illustration”.

 RESPONSE. The figures have been reconstructed and their number increased from2 to 4. A figure depicting the immunomodulating properties of the AMPs (Fig. 2) has been added (replacement of former fig. 1). The structure of AMPs is shown in Fig. 3. The new fig. 4 shows the changes in AMPs associated with perinatal infections and NEC, which suggests a role of AMPs in perinatal infections. The figures 1, 2, and 4 together illustrate the different effects of AMPs on neonatal infections. Adding another table on the role of AMPs on infections will inevitably be a repetition of the same information.

  1. Section 3.3 is interesting, however, the title is too long please rewrite it in a focused and concise way”.

  RESPONSE. The title has been shortened.

  1. …. “human immunodeficiency virus….should it be written in italics? Please confirm and update”.

RESPONSE. We changed this.

  1. Pseudomonas (P) aeruginosa… (P) should be removed and Streptococcus [Str.] pneumoniae…. Please check the whole manuscript for typographical mistakes”.

RESPONSE: the terms pseudomonas, staphylococcus and streptococcus are repeated quite a few times in the text, and therefore we have used abbreviations for these words like the other abbreviations (i.e. we presented the full name at the first appearance). We think that explanation of the initials P, S, Str, etc, will help the reader to discriminate, for example, between staphylococcus (S) and streptococcus (Str) e.t.c. English have been revised by the co-author William Chotas who is an native English speaking Neonatologist. The revised version has been carefully checked for grammar errors.

  1. Number of references is too much. Please reduce”.

 RESPONSE. Dear reviewer, we do agree that the number of references is high. Our initial feelings were that researchers who provided us with valuable information based on their original work deserve to be mentioned. Consequently, this has lead to a high number of references. Nevertheless, the citations have been decreased in the revised version.

Reviewer 2 Report

Authors in this review have tried drawing attention to the different AMPs and their role in neonatal sepsis and NEC. The review is important as they are limited reviews in this context. However, I have a few concerns that need addressing. 

Major Comments:

1. In figure 1, it will be useful to have the names of AMPs (defensins) along with their sources/functions (i.e. Having a figure summary of Table S1). This will end up making the figure more holistic.  Figure 1 can be a representative of section 2.1 and amendments can be made accordingly. A suggestion: authors can include structures of different defensins and then, highlight their function and sources.

2. A general comment to the authors would be to make their review more succinct in context of neonatal infections and sepsis. In many sections, that context was lost.

Authors discuss the role of AMPs in neonatal context in later parts of the review and those are really interesting sections. A strong suggestion would be to restructure, so that under each AMP section, both immunomodulatory and their role in neonatal infections are discussed. If not possible, make a separate section discussing the immunomodulatory roles of AMPs, more broadly rather than individually (as there are enough reviews on immunomodulatory roles of AMPs) and then directly, move on to discuss the role of  AMPs in sepsis and NEC (that's the main theme). As highlighted in Table 2, there are enough evidences to make a comprehensive review. Please shorten the redundant immunomodulatory and AMP definition sections, as it draws the attention away from the main theme of the review and makes it redundant and unnecessarily long.

3. It will be useful to have a summary table outlining the role of each AMP in neonatal infections, sepsis and NEC. 

Minor Comments:

1. Page 2 , the following line 'fact that they constitute small molecules facilitating a rapid approach to the target and diffusion through the microbial cell membrane'  needs restructuring. AMPs are a different class and small molecules are a different drug class. So, the sentence is not making much sense.

2. Page 2, following sentence - 'For certain AMPs, non-membrane mechanisms of antibacterial action have been also described....' is missing a reference.

3. Page 3, the following sentence - 'Human AMPs that are clinically important in early life belong mainly to the defensin and cathelicidin families...' needs restructuring in terms of breaking the sentences for better readability.

4. Amendments in Figure 2 is needed. I would suggest getting rid of the facial expression.

5. Authors need to be careful with the nomenclature of LL-37. In certain cases, it written as lL-37. Please make necessary changes.

Author Response

Dear reviewer,

Thank you for the time you have spent to review our article and for your comments that helped us to improve our presentation. Our responses to your comments are the following:

- “This is a quite long manuscript (57 pages total), though it is written in double space so it can be shorter in the final published version”.

RESPONSE. The main text has been restructured and shortened by 17% according to your suggestion.

- “Despite its length, the narrative is fluid and easy to read. For what I have read, I have not detected any grammatical errors”.

- “The tables show summarized information that helps for a quick view of some medical properties for a specific AMPs. However, Table 2 occupies nine pages and when reading them on my computer screen I had to go back and forth to remember what column was for. For such a long Table 2, I would suggest adding headlines on each page for each column of the table (First author and year, Aim, Results, etc.)”.

 RESPONSE: We have added headlines on each page for each column of the table, and we have tried to make it more concise.

- “There are few figures (only two) and it is mostly narrative so for a moment I though I was like reading a book chapter. The manuscript is clearly divided in sections (six) and subsections but because of its total length, I would also suggest adding a brief index at the beginning of the article. Like any index, it can help to find quickly a section you want to read. It is something common for reviews in other journals. If Journal of Clinical Medicine policies do not require or even do not allow such index, well, no problem”.

RESPONSE: The figures have been reformed and increased from 2 to 4 (plus a Graphical Abstract). Moreover, an index of the sections has been added at the beginning of the article

Reviewer 3 Report

- This is a quite long manuscript (57 pages total), though it is written in double space so it can be shorter in the final published version.

- Despite its length, the narrative is fluid and easy to read. For what I have read, I have not detected any grammatical errors.

- The tables show summarized information that helps for a quick view of some medical properties for a specific AMPs. However, Table 2 occupies nine pages and when reading them on my computer screen I had to go back and forth to remember what column was for. For such a long Table 2, I would suggest adding headlines on each page for each column of the table (First author and year, Aim, Results, etc.)

- There are few figures (only two) and it is mostly narrative so for a moment I though I was like reading a book chapter. The manuscript is clearly divided in sections (six) and subsections but because of its total length, I would also suggest adding a brief index at the beginning of the article. Like any index, it can help to find quickly a section you want to read. It is something common for reviews in other journals. If Journal of Clinical Medicine policies do not require or even do not allow such index, well, no problem.

Author Response

Dear reviewer,

Thank you for the time you have spent to review our article and for your comments that helped us to improve our presentation. Our responses to your comments are the following:

Authors in this review have tried drawing attention to the different AMPs and their role in neonatal sepsis and NEC. The review is important as they are limited reviews in this context. However, I have a few concerns that need addressing. 

Major Comments:

  1. In figure 1, it will be useful to have the names of AMPs (defensins) along with their sources/functions (i.e. Having a figure summary of Table S1). This will end up making the figure more holistic.  Figure 1 can be a representative of section 2.1 and amendments can be made accordingly. A suggestion: authors can include structures of different defensins and then, highlight their function and sources”.

 RESPONSE: In the revised manuscript, the names and structures of the AMPs are shown in Fig. 3 along with their sources in the maternal reproductive system during pregnancy as well as in the fetus and neonate. The Fig. 1 of the first version (now fig. 2), showing the immunomodulating properties of the AMPs, has been fully revised.

  1. A general comment to the authors would be to make their review more succinct in context of neonatal infections and sepsis. In many sections, that context was lost’’.

“Authors discuss the role of AMPs in neonatal context in later parts of the review and those are really interesting sections. A strong suggestion would be to restructure, so that under each AMP section, both immunomodulatory and their role in neonatal infections are discussed. If not possible, make a separate section discussing the immunomodulatory roles of AMPs, more broadly rather than individually (as there are enough reviews on immunomodulatory roles of AMPs) and then directly, move on to discuss the role of  AMPs in sepsis and NEC (that's the main theme). As highlighted in Table 2, there are enough evidences to make a comprehensive review. Please shorten the redundant immunomodulatory and AMP definition sections, as it draws the attention away from the main theme of the review and makes it redundant and unnecessarily long”.

 RESPONSE. In the revised paper, we tried to describe the immunomodulating properties in the text as broadly and succinct as possible, by changing the expressions and omitting some details, while we added a table (Table 1, the modified former supplementary table S1) depicting the properties of AMPs (anti-bacterial, immunomodulating, and other). Moreover, we replaced the former figure 1 with a totally revised illustration (currently fig. 2). However, since the source of AMPs in the reproductive female system during pregnancy and in the fetus neonate are directly related with their importance in neonatal sepsis and NEC, we think that this part of the text could not be omitted or considerably shortened. Our intension was to present (a) the AMP sources in intrauterine environment and the fetus/neonates and (b) their changes in infections and NEC of all the AMPs together in the same section in order to help the reader who wishes to focus on this issue and omit other general information on AMPs. It is understandable that the general information about the AMPs should precede the more specific sections. Moreover, we have added an index of the article contents at the beginning of the main text to help the reader to find easily the part he/she is more interested in.

  1. It will be useful to have a summary table outlining the role of each AMP in neonatal infections, sepsis and NEC”.

 RESPONSE. Dear reviewer, we agree that such a table could be useful. However, the role of each AMP is multiple and complicated due to their multiple effects on bacteria, immune and inflammatory responses, as well on tissue repair and protection of epithelial barrier integrity. In fact, we avoided such a table for two main reasons (a) it will be similar with other figures and tables of the manuscript and (b) it will contain a lot of repetitions since the different AMPs have mostly similar effects and roles in sepsis and NEC.

Minor Comments:

  1. Page 2 , the following line 'fact that they constitute small molecules facilitating a rapid approach to the target and diffusion through the microbial cell membrane'  needs restructuring. AMPs are a different class and small molecules are a different drug class. So, the sentence is not making much sense”.

 RESPONSE. The “small molecules” has been changed to “short amino acid chain”.

  1. Page 2, following sentence - 'For certain AMPs, non-membrane mechanisms of antibacterial action have been also described....' is missing a reference”.

 RESPONSE. References have been added.

  1. Page 3, the following sentence - 'Human AMPs that are clinically important in early life belong mainly to the defensin and cathelicidin families...' needs restructuring in terms of breaking the sentences for better readability”.

 RESPONSE. The sentence has been divided into two sentences “Human AMPs that are clinically important in early life belong mainly to the defensin and cathelicidin families. Additional AMPs not belonging to these families include the antileukoproteases …..”.

  1. Amendments in Figure 2 is needed. I would suggest getting rid of the facial expression”.

 RESPONSE: The former Fig. 2 has been moved to Graphical Abstract and the facial expression has been removed.

  1. Authors need to be careful with the nomenclature of LL-37. In certain cases, it written as lL-37. Please make necessary changes”.

RESPONSE. The text was carefully checked and the necessary changes have been made.

Round 2

Reviewer 1 Report

Authors successfully responded and updated the manuscript as per the reviewer's suggestions and comments. 

Author Response

We thank the Reviewr

Reviewer 2 Report

The authors have diligently addressed majority of the comments raised. However, there are a few minor comments that still need addressing, in order to improve the quality of the paper.

1. The legend of Figure 1 needs to be modified. The mechanism shown is applicable to all AMPs and not solely to human defensins. 

2. Page 8 - 'Several studies demonstrated ..... against pathogens causing in neonatal infections'. The sentence needs to be checked for grammatical accuracy. 

3. Page 9 - 'Moreover, human defensins may induce the release of .....modulate activation of the classical pathway of the complement system'. The sentence needs to be restructured as the later part is not making sense.

4. Page 10 - 'However, the cationic nature of these two peptides ...... antimicrobial mechanism similar to that of other AMPs'. The authors should cite Figure 1 in this sentence.

5. Page 11 - 'Several studies identified the expression of HNP-1 and -2, HD-5 and HD-6, HBD-1 to -4, and LL-37 in human milk and colostrum [36,86–88]. Finally, AMPs produced in the gut following proteolysis and fermentation of milk proteins (e.g. casein, a-lactalbumin, lactoferrin, and lysozyme) play a very important role in intestinal host defense and integrity of the gut epithelial barrier (Table 2)' . The two sentences need restructuring. The main focus should be on Milk-derived AMPs. HNP1, HD-5..and others that have been highlighted before, and can be included as a later sentence.

6. Authors can move table 3 to Supplementary, as this is making the review unnecessarily long.

7. Authors need to shorten the following sections and their sub-sections - 3.2, 3.3, 3.4 and 5. Several sentences are discussed at length and also many parts seemed redundant. In particular, the section 5 should be limited to only a single paragraph and not too long, so that the final message comes across.

8. As a general comment, authors should be careful about grammatical inaccuracy throughout the article. Also, if possible, try shortening wherever possible, into concise sentences. 

Author Response

Author's Reply to the Reviewer’s 2 Report

Dear Reviewer,

Thank you for your suggestions that helped to improve of our paper presentation. We have made the vast majority of the suggested changes. However, we did not conform to certain suggestions for reasons that are explained in our responses bellow.

Our responses to the Reviewer’s comment are the follows:

  1. COMMENT: The legend of Figure 1 needs to be modified. The mechanism shown is applicable to all AMPs and not solely to human defensins. 

RESPONSE: The legend has been modified.

  1. COMMENT: Page 8 - 'Several studies demonstrated ..... against pathogens causing in neonatal infections'. The sentence needs to be checked for grammatical accuracy. 

RESPONSE: The sentence has been rephrased as follows “Several studies have shown that  the LL-37 acts synergistically with HNP-1, HBD-2 and HBD-3 against pathogens that cause neonatal infections”

  1. COMMENT: Page 9 - 'Moreover, human defensins may induce the release of .....modulate activation of the classical pathway of the complement system'. The sentence needs to be restructured as the later part is not making sense.

RESPONSE: The sentence has been rephrased to “Moreover, human defensins have been reported to induce the release of histamine and prostaglandin D2 (PGD2) through activation of mast cells  and inhibit activation of the classical complement pathway”.

  1. COMMENT: Page 10 - 'However, the cationic nature of these two peptides ...... antimicrobial mechanism similar to that of other AMPs'. The authors should cite Figure 1 in this sentence.

RESPONSE: the “Fig.1” has been added.

  1. COMMENT: Page 11 - 'Several studies identified the expression of HNP-1 and -2, HD-5 and HD-6, HBD-1 to -4, and LL-37 in human milk and colostrum. Finally, AMPs produced in the gut following proteolysis and fermentation of milk proteins (e.g. casein, a-lactalbumin, lactoferrin, and lysozyme) play a very important role in intestinal host defense and integrity of the gut epithelial barrier (Table 2). The two sentences need restructuring. The main focus should be on Milk-derived AMPs. HNP1, HD-5..and others that have been highlighted before, and can be included as a later sentence.

RESPONSE: This part has been restructured to highlight the AMPs derived from the human milk proteins. We hope that its current form is compatible with the reviewer’s suggestion : “ Several AMPs are produced in the gut following proteolysis and fermentation of milk proteins (e.g. casein, a-lactalbumin, lactoferrin, and lysozyme, Table 2). Additionally, HNP-1 and -2, HD-5 and HD-6, HBD-1 to -4, and LL-37 have been identified in human milk and colostrum”.

  1. COMMENT: Authors can move table 3 to Supplementary, as this is making the review unnecessarily long.

RESPONSE: Dear reviewer, we hesitated to move the Table 3 to the supplementary files, because this change would cause a huge readjustment of the citations. We were concerned that this might be a source of errors in the in-,text citations and especially in the figure 4, despite the fact the ZOTERO was used for the citations. Additionally, it is not uncommon for the reviews to include in the main text big tables of the relative studies. Such tables are usually very welcome by the readers, because they provide them with a comprehensive picture of the issue at a glance. Additionally, to our experience some readers are not willing to go through the supplementary files. Regarding the length of the manuscript, the main text has been considerably shortened to 8188 words (from 9955 words of the first version, i.e. a reduction of 18%). This length cannot be considered as too long for a review (many journals accept reviews of 12 000 words or even more). In any case, the added list of contents can help the reader to choose the section he/she is most interested in. In addition, compared to the first version of our manuscript, the Table 3 has been significantly shortened and data are presented more concisely. 

  1. COMMENT: Authors need to shorten the following sections and their sub-sections - 3.2, 3.3, 3.4 and 5. Several sentences are discussed at length and also many parts seemed redundant. In particular, the section 5 should be limited to only a single paragraph and not too long, so that the final message comes across.

RESPONSE: Dear Reviewer, thank you for your suggestion. We have shortened  quite a few long sentences. However, regarding the suggestion for further shortening certain sections of the main text, we regret to say that this suggestion cannot be totally applied to the specific sections as it opposes the aim of our article. Undoubtedly, there are excellent published articles about this issue, which however, focus on isolated AMPs and their functions and potential role in the either the chorioamnionitis, or the fetus, or the neonate. In addition, most of the few reviews regarding the AMPs in the fetus and neonate do not present details that could be interesting for the health professionals involved in the care of the fetus and neonate. Our wish was to present a comprehensive and collected review of the currently existing data regarding the developmental changes and the sources of AMPs in the fetus and neonate as well as their changes associated with perinatal infections and NEC. In this context, the sections 3.2, 3.3, and 3.4 (along with the section 4) are very important parts of our review. Nevertheless, we tried to reduce the sections 3.2  as much as it was possible to be shorten without losing important information.  The sections 3.3. and 3.4.,  actually, present a summary of the available relevant information, and they cannot be further shortened. Moreover, to our opinion, the section 5 should not be limited to a single paragraph for two main reasons; (a) a section regarding the future perspectives is usually an important part of an “update” and (b) this section covers two of the aims of our review (that have been announced at the end of the introduction): “In addition, we present emerging data on the potential utility of AMPs as diagnostic and predictive biomarkers of perinatal infections and NEC. Finally, given the increase in antimicrobial resistance in neonatal intensive care units, we discuss briefly the challenge of identifying and developing new synthetic analogs of AMPs as an alternative or adjunct to antibiotic therapy”. To our opinion, the majority of health professionals involved in the care of the fetus and neonate would like to be comprehensively informed about the available data supporting the potential role of AMPs in the perinatal period. For all these reasons, we regret to say that this suggestion practically eliminates our review, so that we cannot further conform to it.

  1. COMMENT: As a general comment, authors should be careful about grammatical inaccuracy throughout the article. Also, if possible, try shortening wherever possible, into concise sentences. 

RESPONSE: Several sentences have been rephrased and shortened and the text was checked again for grammatical inaccuracy, as it is obvious in the submitted manuscript with tracked changes.Bottom of Form
